# Contents of Macro- and Microelements in Blood Serum and Breast Muscle of Turkey Subjected to Pre-Slaughter Transport for Various Distances

**DOI:** 10.3390/ani14152242

**Published:** 2024-08-01

**Authors:** Janusz F. Pomianowski, Anna Wójcik, Janina Sowińska, Dorota Witkowska, Daria Murawska, Tomasz Mituniewicz

**Affiliations:** 1Department of Food Microbiology, Meat Technology and Chemistry, Faculty of Food Sciences, University of Warmia and Mazury in Olsztyn, Plac Cieszyński 1, 10-726 Olsztyn, Poland; 2Department of Animal Welfare and Research, Faculty of Animal Bioengineering, University of Warmia and Mazury in Olsztyn, Oczapowskiego 5, 10-718 Olsztyn, Poland; awojcik@uwm.edu.pl (A.W.); janina.sowinska@uwm.edu.pl (J.S.); dorota.witkowska@uwm.edu.pl (D.W.); daria.murawska@uwm.edu.pl (D.M.); t.mituniewicz@uwm.edu.pl (T.M.)

**Keywords:** turkeys, stress, pre-slaughter handling, mineral components, macro- and microelements

## Abstract

**Simple Summary:**

Simple Summary: Poultry can be exposed to a number of adverse stimuli during the pre-slaughter handling. Noise, movement, lack of access to water and feed, change in habitat, disruption of flock hierarchy or environmental conditions can lead to adverse changes in the bird’s body. A significant increase in these factors can cause disturbances in the body’s homeostasis. These changes may be reflected in meat quality. Therefore, there is a need to assess the effect of pre-slaughter turnover on meat quality. In this study, the effects of pre-slaughter treatments on changes in macro- and micronutrient content were analyzed.

**Abstract:**

In this study, the effect of pre-slaughter handling on the content of macro- and micronutrients in blood serum and in the breast muscle of turkeys was assessed. Four different variants of pre-slaughter handling were used in the research: no transport (N-T), transport for a distance of 100 km (T-100), transport for a distance of 200 km (T-200), and transport for a distance of 300 km (T-300). In each of them, 30 female and 30 male turkeys were used. Blood was collected from the birds before slaughter, and samples of the pectoral muscle were collected after slaughter. In the obtained biological material, the content of Mg, P, Ca, Fe, Na and K was analyzed. On the basis of the obtained research results, it can be concluded that the pre-slaughter handling negatively affects the content of macro- and microelements both in the blood serum and in the pectoral muscles of experimental turkeys. Additionally, differences due to the sex of birds were observed.

## 1. Introduction

Poultry meat is largely sourced from farmed poultry species. Due to the conditions of intensive poultry production, birds are transported at least twice during their lifetime. The first is from the hatchery to the farm, and the second is from the farm to the slaughterhouse [1,2,3]. The most stressful part of the pre-slaughter handling of birds is the transport process [4]. During transport, poultry are exposed to a number of adverse stimuli acting in combination: sudden movement, vehicle motion, acceleration, vibration, shock, noise, lack of food and water, disruption of flock hierarchy and changes in the environment [1,3,4,5]. Similar adverse processes in relation to pre-slaughter handling that may have an impact on meat quality have also been observed in pigs [6,7]. As a result of stressors to the body, macro- and micronutrients may be transformed and lost first in blood serum and then in tissues and muscles during the pre-slaughter handling of poultry [8,9,10,11]. Macro- and micronutrient deficiencies may result in reduced meat quality.

Issues related to the levels of macro- and micronutrients in poultry meat have been presented in many studies but mainly in the context of the influence of diet on their levels in meat [10,12,13,14,15,16,17,18,19]. However, few studies have investigated the effect of stress factors on the levels of macro- and micronutrients in poultry meat [20,21,22]. Therefore, the aim of this study was to determine the levels of Ca, Mg, Na, K, P, and Fe in the blood serum and breast muscle of turkeys subjected to different variants of pre-slaughter transport at different distances during the summer: no transport (NT), with a transport of 100 km (T-100), 200 km (T-200) and 300 km (T-300).

## 2. Materials and Methods

### 2.1. Experimental Birds, Pre-Slaughter Handling

The study of the influence to the adverse effects of pre-slaughter handling on the macro and micronutrient content of turkeys was carried out separately for male and female turkeys because, according to the production technology, these birds are slaughtered at different ages: females at 15–16 weeks of age and males at 18–22 weeks of age. The female and male turkeys used in this study were from a single flock from a commercial farm. For the experiment, females and males were purchased at different ages: females at 15–16 weeks of rearing and males at 18–22 weeks of rearing. Subsequently, male turkeys were purchased from the same flock at 17 weeks of age, which were also kept for 3 weeks before the experiment was carried out. The three-week rearing period was carried out in the animal laboratory of the Department of Animal and Environmental Hygiene in accordance with the technological requirements for this group of poultry under controlled environmental conditions and with the same care and feeding conditions for all groups of poultry. Female and male turkeys were fed according to poultry feeding standards [23] with standard industrial dry mix for turkeys at full ration.

Different distances of transport of turkeys before slaughter were used:-N-T—no transport (30 males, 30 females);-T-100—transport for a distance of 100 km (30 males, 30 females);-T-200—transport for a distance of 200 km (30 males, 30 females);-T-300—transport for a distance of 300 km (30 males, 30 females).

The complete pre-slaughter process included the following steps: catching, weighing, loading into containers, transport and waiting for slaughter, unloading and weighing. In contrast, the pre-slaughter handling without transport (group N-T) consisted of catching, weighing, loading into containers, waiting for slaughter and unloading.

On the day before slaughter, at 9 p.m., the feed mixes were taken, and the birds had access to water only. On the day of slaughter, the turkeys were weighed at 7:00 a.m. and randomly divided into subgroups based on the length of transport.

Turkeys from the N-T group were slaughtered after 8:00 a.m. The transport of the remaining birds started at 8:00 a.m. and was carried out in a vehicle adapted for poultry transport. After driving 100 km, the vehicle arrived at the laboratory at around 9:30 a.m., where the T-100 birds were unloaded. The vehicle then drove another 100 km and returned to the laboratory at around 11:00 a.m., where the T-200 birds were unloaded. The procedure was then repeated for the T-300 turkeys, which were unloaded at around 1 p.m.

Each time the vehicle followed the same route at an average speed of 60–70 km/h.

### 2.2. Sample Collection and Chemical Analyses

After transport, the birds were weighed, blood samples were collected from the wing vein and then slaughtered. Slaughter of the experimental poultry was carried out under laboratory conditions in accordance with all procedures used for birds at the slaughterhouse (Local Ethics Committee authorization no. 77/2008 of 29 October 2008). After cooling, samples (approximately 150 g) were taken from the right breast muscle. Laboratory tests of blood serum were performed in the certified Analytical Laboratory of Olsztyn City Hospital (ISO 9001:2000 Certificate No. 134B-2002-AQ-GDA-RvA [24]).

Tests of blood hematological indices of turkeys were carried out in the Veterinary Diagnostic Laboratory of Dr. n. wet. Roman Jędryczko in Gietrzwałd. A leukogram was made on whole blood using the Shilling method, and blood smears were stained with May–Grünwald–Giemsa solution using the Pappenheim method. The numerical ratio of heterophilic to lymphocytic cells (H:L) was calculated from the leukogram.

Corticosterone concentration was determined by radioimmunoassay after extraction from plasma (with ethyl acetate) using specific anti-corticosterone antibodies [25] at the Department of Animal Physiology, Faculty of Biology and Biotechnology, University of Warmia and Mazury in Olsztyn.

Calcium (Ca), magnesium (Mg), phosphorus (P), sodium (Na), potassium (K) and iron (Fe) were determined in the breast muscles. The biological material was wet mineralized, and the ash was hot dissolved in 1 M nitric acid solution (Merc Rahway, NJ USA). The resulting mineralizations were analyses for Ca, Mg and Fe by atomic absorption flame spectrometry. A Unicam 939 Solar (Labexchange, Burladingen, Germany) atomic absorption spectrometer equipped with an Optimus database, background correction (deuterium discharge lamp), appropriate cathode ray tubes, a Unicam GF 90 graphite furnace and an FS 90 autosampler were used for the determinations. Na and K contents were determined by the emission method (acetylene–air flame). The tests were conducted using a Pye Unicam SP 2900 atomic absorption spectrophotometer (Pye, Cambridge, UK) [26]. Phosphorus content was determined by the colorimetric method [27]. Absorbance was measured with a VIS 6000 (A.KRÜSS Optronic, Hamburg, Germany) at λ = 610 nm.

### 2.3. Statistical Analysis

The data collected for levels of macro- and microelements in the blood serum and breast muscle of the turkeys were processed by a one-way analysis of variance in orthogonal design (ANOVA). The statistical analysis of data involved the determination of arithmetic means and standard deviations (SD). The significance of differences between mean values computed for particular levels of experimental factors was determined by Duncan’s test. All calculations were made with Statistica 13.1 PL software.

## 3. Results

The recorded mean body weight of the female turkeys before transport was 10.36–11.26 kg and was statistically significantly different (*p* ≤ 0.01) between the experimental groups (Table 1). The body weight of male turkeys varied within a narrow range from 19.19 to 20.72 kg. To determine the level of stress in the examined birds, two indices usually used in the animals were evaluated: the H:L ratio and corticosterone levels. In both females (1.05–1.33) and males (1.56–1.88), the H:L ratio was similar among the groups. Serum corticosterone levels in female turkeys were highly statistically significantly (*p* ≤ 0.01) in the 100, 200 and 300 km transport groups (10.22 ng/mL; 10.76 ng/mL; 9.71 ng/mL, respectively) compared to the non-transport group (7.56 ng/mL). In contrast, serum corticosterone levels in male turkeys were statistically highly statistically (*p* ≤ 0.01) in the 200 km (7.45 ng/mL) and 300 km (8.68 ng/mL) transported groups compared to the non-transported group (4.44 ng/mL) and the 100 km transported group (4.10 ng/mL).

When considering the effect of pre-slaughter handling on the levels of macro- and microelements in the blood of female turkeys, the influence of this factor was statistically confirmed for magnesium, phosphorus, iron and sodium (Table 2). Calcium and potassium levels in the blood serum of female turkeys were at similar levels (*p* ≥ 0.05). The lowest levels of magnesium, iron and sodium were found in the group of female turkeys transported 300 km. Magnesium level in this group (T-300, 0.97 mmol/L) was statistically lower compared to turkeys transported 200 km (1.07 mmol/L; *p* ≤ 0.05) and did not differ from the trait values in the N-T and T-100 groups (*p* ≥ 0.05). Iron levels (19.38 µmol/L) were lowest compared to the T-100 group (27.52 µmol/L, *p* ≤ 0.01) and lower (*p* ≤ 0.05) compared to non-transported female turkeys (25.46 µmol/L) and turkeys transported 200 km (25.22 µmol/L). The effect of transport on serum sodium content was confirmed. The lowest sodium content was found in birds from the T-300 group (155.60 mmol/L). This value differed highly significantly from the T-100 group (158.53 mmol/L, *p* ≤ 0.01) and significantly from the N-T group (157.90 mmol/L, *p* ≤ 0.05). For phosphorus, the lowest levels were found in the 100 km and 300 km transported turkey groups. The value of the trait in the T-100 group differed highly significantly compared to the value in the N-T group (1.79 mmol/L, *p* ≤ 0.01) but did not differ compared to the T-200 and T-300 groups (*p* ≥ 0.05). In addition, a statistically significant difference was found between the N-T and T-300 groups (*p* ≤ 0.05).

A different trend was observed in the blood serum results of the examined male turkeys (Table 3). Although the determined values were highest in the T-300 group, the effect of transport on magnesium and sodium was not statistically confirmed. The effect of transport on serum levels of phosphorus, calcium and potassium in males was statistically confirmed (*p* ≤ 0.01), but in the case of magnesium, iron and sodium, the results did not indicate an effect of this factor on their serum content (*p* ≥ 0.05). In the case of calcium, the lowest values were found in the T-100 and T-200 groups (2.61 and 2.61 mmol/L), and the highest values were found in the N-T and T-200 groups (2.76 and 2.73 mmol/L). And this showed that the variability between the NT, T-100 and T-200 groups was highly statistically significant (*p* ≤ 0.01) and between the T-100 and T-200 and T-300 groups was statistically significant (*p* ≤ 0.05). Among the serum analyses of males, the blood of birds in the T-300 group contained the highest levels of potassium (2.71 mmol/L) and phosphorus (1.65 mmol/L, *p* ≤ 0.01). The phosphorus content in the N-T (1.38 mmol/L), T-100 (1.35 mmol/L) and T-200 (1.33 mmol/L) groups was similar (*p* ≥ 0.05). Also for potassium, there were no differences between groups N-T (1.93 mmol/L), T-100 (1.98 mmol/L) and T-200 (1.80 mmol/L, *p* ≥ 0.05).

Table 4 shows the macro- and micronutrient content of turkey female breast muscle. Among the tested elements, the highest content was found in the case of sodium. Its amount ranged from 288.0 mg/100 g (T-300) to 313.7 mg/100 g (T-200). This differentiation was statistically significant (*p* ≤ 0.05; *p* ≤ 0.01). The next element in terms of content in the breast muscles of female turkeys was potassium. The amounts of this ingredient ranged from 256.0 mg/100 g (T-200) to 271.0 mg/100 g (T-100). Such a large variation was not statistically significant. The determined contents of the remaining examined elements, despite the differences between the individual groups of birds, did not show statistically significant differences (*p* ≤ 0.05; *p* ≤ 0.01). It can be added here that the meat of the T-100 group contained the most magnesium (0.368 mg/100 g), phosphorus (29.90 mg/100 g) and calcium (15.70 mg/100 g), while the most iron contained the meat of the birds from the T-200 group (49.93 mg/100 g).

In the breast meat of male turkeys, similar trends were found as in the female turkeys (Table 5). As in the muscles of female turkeys, the element with the highest content was sodium. The lowest levels were found in the muscles of birds from the T-200 group (275.83 mg/100 g) and the highest in the N-T group (290.00 mg/100 g). In this case, the difference was statistically significant (*p* ≤ 0.05). The next element in terms of quantity was potassium. Its amounts ranged from 255.33 mg/100 g (T-100) to 265.17 mg/100 g (T-300). Statistical analysis showed no significant variation (*p* ≤ 0.05; *p* ≤ 0.01). Determination of iron, phosphorus and calcium quantity in the meat was insignificant. However, in the magnesium content, statistical analysis showed the high significance (*p* < 0.01) of the T-200 group (0.372 mg/100 g) to the T-100 and N-T groups (0.327 and 0.328 mg/100 g, respectively).

## 4. Discussion

Minerals found in animal cells and tissues have a variety of functions in the organism [11,12,16,19,28,29,30,31,32]. They are found in tissues and cell fluids in the form of the electrolytes: sodium, potassium and calcium, and they influence osmotic pressure change and acid–base balance in the body. Potassium is responsible for the proper functioning of nerve and muscle tissue. Magnesium plays an important role in activating many of the enzymes required for the metabolism of carbohydrates, phosphorus and calcium, in the contraction of muscles, including the heart muscle, and in maintaining the normal rhythm of the heart. It influences neuromuscular excitability, stimulates the body’s defense mechanisms, and has a calming effect. Iron is a component of many enzymes and metalloproteinic compounds involved in oxidation and reduction processes. Iron bound in the blood as hemoglobin is involved in the transport of oxygen from the lungs to the tissues. In muscles, iron is part of myoglobin, which is the red muscle pigment that takes oxygen from red blood cells and uses it for muscle work. Phosphorus is involved in the transmission of nerve impulses and is a building block of cell membranes and soft tissues.

Poultry are constantly exposed to a number of stressors, which may last for a few hours (e.g., capture, caging and transport) or throughout the rearing period (e.g., heat stress, immunological problems) [33,34,35]. Stress induces several physiological responses, including an increase in the ratio of heterophils (H) to lymphocytes (L) in the blood [36,37,38,39]. Siegel and Gross [32] report that H:L ratios are approximately 0.2–0.3; values of 0.4–0.5 and above 0.8 characterize low, optimal and high-intensity stress. A very low level of environmental stimulation, expressed by an H:L ratio between 0.2 and 0.3, indicates the need to stimulate the body to function at an appropriate physiological level. In contrast, an H:L ratio above 1.3 may indicate the onset of a stress-related disease state. The results of our own study showed an increase in the H:L ratio in the blood serum of turkeys, indicating that birds in all experimental groups were responding to stressors occurring during the pre-slaughter turnover period.

The transport of poultry, the method of loading and finally the entire pre-slaughter turnover cause the occurrence of certain stress factors, which also result in increased production of adrenal cortex hormones and an increase in the level of corticosterone in the blood serum [40,41,42,43,44]. Duncan [4] showed that broiler chickens placed in containers and then transported for 40 min had higher serum corticosterone levels than birds that were only placed in containers but not transported. Cashman et al. [45] also reported in their study that transport was more stressful than just catching and loading the poultry. Di Martino et al. [46], who studied the behavior of turkeys during 86 km of transport in cages of two different heights (conventional—38.5 cm and experimental—77 cm), showed that hematological and serum biochemical parameters were not significantly affected regardless of cage height. However, the values of these indices indicated the presence of stress during transport. Serum H:L ratios in turkeys ranged from 1.4 to 2.1, while corticosterone levels ranged from 11.7 to 16.2 ng/mL. Our own research showed an increase in corticosterone levels in female turkeys in all transported groups and in male turkeys transported 200 km and 300 km compared to the non-transported group (both males and females) and the transported group for male turkeys. The H:L and corticosterone results indicate that there was a physiological response in the turkeys, which is indicative of the stress load that occurs during pre-slaughter handling. All transported female and male turkeys transported over 200 and 300 km showed the strong response. Given that the non-transported birds also showed H:L ratios and corticosterone levels indicative of the onset of stress, even activities such as catching, weighing, loading into cages and moving to the laboratory section of the abattoir are a heavy burden for them.

There are few studies indicating how macro- and micronutrients in blood serum and muscle behave during stress in pre-slaughter poultry. Stress factors such as noise, movement, high ambient temperature or transport have been shown to affect the differentiation of macro- and micronutrients in blood serum [8,47,48] or muscle [10,11,14]. According to Ognik et al. [49], the demand for minerals increases significantly in animals under stress conditions. The reason for this phenomenon is their use in a number of reactions related to energy production or release as well as in antioxidant defense mechanisms. The consequence of macro- and micronutrient deficiencies can be a reduction in the quality of poultry meat. In our study, turkeys transported 300 km had the highest levels of magnesium, phosphorus, calcium and potassium compared to the other groups. The other tested elements were at similar levels. In female turkey breast muscle, except for sodium, and in male turkeys, except for magnesium and sodium, macro- and micronutrient levels were also at similar levels. Sodium levels in female turkeys from the non-transported and 300 km transported groups were lower than in the other two groups. In contrast, sodium levels in male turkeys in the transported groups were lower than in the non-transported group. The levels found in our study are similar to the levels of macro- and micronutrients in blood serum or breast muscle found by other authors [18,43,49,50,51].

The stress caused by the pre-slaughter handling stimulates the birds defense mechanisms. Adrenaline is released, preparing the animal for fight or flight. Metabolism also increases, leading to an increased demand for energy, resulting in the destruction of energy resources (carbohydrates, fats and proteins). The first symptom of such activity is dehydration [3,52,53]. Body water loss in birds can reach up to 5% of body weight after several hours of transport. In addition, weight loss during the transport of chickens to the slaughterhouse affects slaughter performance and the quality of the meat and final meat product [54,55]. The increasing effect of heat stress on the bird’s body significantly increases weight loss. This is confirmed by a study [54] in which greater weight loss was observed in chickens kept at an elevated ambient temperature (29 °C) prior to slaughter compared to birds kept under optimal thermal conditions (18 °C).

The fluctuations in serum and breast muscle macro- and micronutrient levels observed in our study may therefore be related to water loss from the body during transport at the summer period. The water losses were probably due to an increase in temperature and relative humidity in the vehicles used to transport the birds [9,56].

This has the effect of increasing the activity of the thermoregulatory system to prevent overheating of the birds, which removes water from the body by increasing heat loss through evaporation from the respiratory tract [11].

During transport, when there is no opportunity to water the birds and replenish water losses, dehydration may occur, resulting in increased blood solids and increased muscle dry weight, leading to higher blood serum levels of some elements (female turkeys—iron and sodium; male turkeys—potassium) during 100 km of transport. On the other hand, the results suggest that there may have been a loss of macro- and micronutrients due to the evaporation of water from the birds bodies by increased respiration during transport over distances of 200 and 300 km.

To reduce the impact of stress on animals, it is important to maintain their welfare. This is important for both farmers and veterinarians to be able to maintain good health for all types of livestock. To this end, both new developments in teaching methodology and technical means, including social media, should be used [57,58].

## 5. Conclusions

The marked fluctuations in the levels of macro- and micronutrients in the blood serum and breast muscle of turkeys may indicate that the birds’ bodies tried to maintain stable levels of these components despite the stress factors present. On the other hand, a decrease in the mineral content of blood serum and breast muscle with an increase in transport distance up to 300 km may indicate a depletion of macro- and micronutrients due to dehydration and the body’s need for self-defense during periods of continuously increasing stress factors. Thus, based on the results of the study, it can be concluded that prolonged pre-slaughter handling negatively affected the macro- and micronutrient content of both blood serum and breast muscle of the experimental turkeys.

## Figures and Tables

**Table 1 animals-14-02242-t001:** Body weight, blood heterophils/lymphocytes ratio and serum corticosterone levels of turkeys before pre-slaughter handling (mean ± SD).

Item	Statistical Measure	Pre-Slaughter Handling			
		N-T	T-100	T-200	T-300
Female					
Body weight before transport (kg)	$\bar{x}$	10.49 ^B^	10.36 ^B^	10.90 ^C^	11.26 ^A^
	SD s	0.27	0.48	0.32	0.67
H:L (1/1)	$\bar{x}$	1.05	1.09	1.33	1.12
	SD	0.53	0.28	0.47	0.30
Corticosterone (ng/mL)	$\bar{x}$	7.56 ^A^	10.22 ^B^	10.76 ^B^	9.71 ^B^
	SD	2.68	3.98	3.61	3.32
Male					
Body weight before transport (kg)	$\bar{x}$	19.19	20.72	20.22	20.35
	SD	1.95	1.40	1.21	1.79
H:L (1/1)	$\bar{x}$	1.73	1.56	1.81	1.88
	SD	0.57	0.91	0.59	0.43
Corticosterone (ng/mL)	$\bar{x}$	4.44 ^A^	4.10 ^A^	7.45 ^B^	8.68 ^B^
	SD	1.99	2.09	1.63	2.38

Values denoted with different letters are significantly different: ^A^, ^B^, ^C^—at a level of *p* ≤ 0.01; N-T—no transport; T-100—transport for a distance of 100 km; T-200—transport for a distance of 200 km; T-300—transport for a distance of 300 km.

**Table 2 animals-14-02242-t002:** Contents of macro- and microelements in female turkey blood serum (mean ± SD).

Macro- and Microelements	Statistical Measure	Pre-Slaughter Handling			
		N-T	T-100	T-200	T-300
Magnesium (mmol/L)	$\bar{x}$	1.02 ^ab^	1.04 ^ab^	1.07 ^a^	0.97 ^b^
	SD	0.15	0.13	0.16	0.11
Phosphorus (mmol/L)	$\bar{x}$	1.79 ^Aa^	1.54 ^B^	1.70 ^ab^	1.57 ^b^
	SD	0.36	0.37	0.33	0.28
Calcium (mmol/L)	$\bar{x}$	2.95	2.95	2.99	2.79
	SD	0.37	0.26	0.42	0.17
Iron (µmol/L)	$\bar{x}$	25.46 ^b^	27.52 ^B^	25.22 ^b^	19.38 ^Aa^
	SD	12.42	9.33	13.38	7.77
Sodium (mmol/L)	$\bar{x}$	157.90 ^a^	158.53 ^A^	157.40 ^ab^	155.60 ^Bb^
	SD	3.52	2.39	6.11	2.72
Potassium (mmol/L)	$\bar{x}$	2.22	2.15	2.27	2.38
	SD	0.41	0.32	0.92	0.61

Values denoted with different letters are significantly different: ^A^, ^B^ at a level of *p* ≤ 0.01; ^a^, ^b^ at a level of *p* ≤ 0.05; N-T—no transport; T-100—transport for a distance of 100 km; T-200—transport for a distance of 200 km; T-300—transport for a distance of 300 km.

**Table 3 animals-14-02242-t003:** Contents of macro- and microelements in male turkey blood serum (mean ± SD).

Macro- and Microelements	Statistical Measure	Pre-Slaughter Handling			
		N-T	T-100	T-200	T-300
Magnesium (mmol/L)	$\bar{x}$	0.991	1.020	0.935	1.102
	SD	0.6	0.18	0.9	0.18
Phosphorus (mmol/L)	$\bar{x}$	1.38 ^B^	1.35 ^B^	1.33 ^B^	1.65 ^A^
	SD	0.23	0.23	0.18	0.30
Calcium (mmol/L)	$\bar{x}$	2.76 ^A^	2.61 ^B^	2.61 ^B^	2.73 ^ab^
	SD	0.14	0.12	0.10	0.12
Iron (µmol/L)	$\bar{x}$	14.03	12.61	9.68	12.06
	SD	5.37	4.45	4.12	4.60
Sodium (mmol/L)	$\bar{x}$	159.67	158.33	158.53	160.87
	SD	2.99	4.43	2.80	4.44
Potassium (mmol/L)	$\bar{x}$	1.93 ^B^	1.98 ^B^	1.80 ^B^	2.71 ^A^
	SD	0.16	0.44	0.56	0.99

Values denoted with different letters are significantly different: ^A^, ^B^ at a level of *p* ≤ 0.01; ^a^, ^b^ at a level of *p* ≤ 0.05; N-T—no transport; T-100—transport for a distance of 100 km; T-200—transport for a distance of 200 km; T-300—transport for a distance of 300 km.

**Table 4 animals-14-02242-t004:** Contents of macro- and microelements in female turkey breast muscle (mean ± SD).

Macro- and Microelements	Statistical Measure	Pre-Slaughter Handling			
		N-T	T-100	T-200	T-300
Magnesium (mg/100 g)	$\bar{x}$	0.337	0.368	0.332	0.333
	SD	0.07	0.12	0.05	0.06
Phosphorus (mg/100 g)	$\bar{x}$	29.70	29.90	28.08	29.43
	SD	0.81	1.08	1.81	1.61
Calcium (mg/100 g)	$\bar{x}$	15.42	15.70	15.38	15.50
	SD	0.53	0.66	0.58	0.99
Iron (mg/100 g)	$\bar{x}$	46.62	43.45	49.93	48.45
	SD	5.33	13.45	8.09	5.11
Sodium (mg/100 g)	$\bar{x}$	296.7 ^B^	299.7 ^b^	313.7 ^Aa^	288.0 ^Ba^
	SD	11.83	9.22	29.92	14.38
Potassium (mg/100 g)	$\bar{x}$	266.7	271.0	256.0	257.8
	SD	5.72	12.99	20.62	13.54

Values denoted with different letters are significantly different: ^A^, ^B^ at a level of *p* ≤ 0.01; ^a^, ^b^ at a level of *p* ≤ 0.05; N-T—no transport; T-100—transport for a distance of 100 km; T-200—transport for a distance of 200 km; T-300—transport for a distance of 300 km.

**Table 5 animals-14-02242-t005:** Contents of macro- and microelements in male turkey breast muscle (mean ± SD).

Macro- and Microelements	Statistical Measure	Pre-Slaughter Handling			
		N-T	T-100	T-200	T-300
Magnesium (mg/100 g)	$\bar{x}$	0.328 ^B^	0.327 ^B^	0.372 ^A^	0.345 ^AB^
	SD	0.02	0.02	0.03	0.03
Phosphorus (mg/100 g)	$\bar{x}$	30.75	30.45	31.23	31.62
	SD	0.52	0.43	0.73	1.35
Calcium (mg/100 g)	$\bar{x}$	15.32	15.25	15.33	15.40
	SD	0.32	0.36	0.36	1.17
Iron (mg/100 g)	$\bar{x}$	67.02	60.42	62.57	64.02
	SD	6.14	4.62	3.86	3.33
Sodium (mg/100 g)	$\bar{x}$	290.00 ^a^	280.00 ^b^	275.83 ^b^	277.17 ^b^
	SD	7.16	9.76	9.54	9.70
Potassium (mg/100 g)	$\bar{x}$	257.17	255.33	261.50	265.17
	SD	7.83	5.61	8.55	18.44

Values denoted with different letters are significantly different: ^A^, ^B^ at a level of *p* ≤ 0.01; ^a^, ^b^ at a level of *p* ≤ 0.05; N-T—no transport; T-100—transport for a distance of 100 km; T-200—transport for a distance of 200 km; T-300—transport for a distance of 300 km.

## Data Availability

Data available upon request.

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
