# Peer review of "Contents of Macro- and Microelements in Blood Serum and Breast Muscle of Turkey Subjected to Pre-Slaughter Transport for Various Distances"

_animals, 2024, doi:10.3390/ani14152242_

Round 1

Reviewer 1 Report (Previous Reviewer 1)

Comments and Suggestions for Authors

Some improving in References was done, but description of results is not highly scientific, I suggest refuse the manuscript by this journal.

Comments on the Quality of English Language

Author Response

Reviewer 2 Report (Previous Reviewer 2)

Comments and Suggestions for Authors

The authors have made improvements based on the reviewer inquiries. I have no further comment.

Author Response

Thank you very much for your review and comments on the paper.

Reviewer 3 Report (Previous Reviewer 3)

Comments and Suggestions for Authors

Authors made some corrections in their article, however, there is a need for further improvement

The use of superscripts is still not appropriate in Tables 2, 3, 5 and 5 (superscripts are necessary for the whole row)

L87-88: "withdrawn away"?

L143: "heterophilia"?

L150: The female turkeys!

L152: Male turkeys!

L155-156: " In both female (1.05-1.33) and male (1.56-1.88) the H:L ratio was similar among groups."

L157: "in female turkeys"

L150: "in male turkeys"

L284: "Occurring"?

L287-290: The phrase "plays an important role" is used twice. please rephrase

L285-297: Please add references

L311: "carries"?

L322-324: You refer to the same study in the previous lines (L317-322)

Comments on the Quality of English Language

Extensive editing of English language required

Author Response

Thank you very much for your review and comments on the paper.

Comments 1

The use of superscripts is still not appropriate in Tables 2, 3, 5 and 5 (superscripts are necessary for the whole row)

Response 1:

We disagree with this comment.

It is common practice that only statistically different values are marked. Marking all data will significantly darken the picture and the table will be unreadable

Comments 2

L87-88: "withdrawn away"?

L143: "heterophilia"?

L150: The female turkeys!

L152: Male turkeys!

L155-156: " In both female (1.05-1.33) and male (1.56-1.88) the H:L ratio was similar among groups."  

L157: "in female turkeys"           

L150: "in male turkeys"

L284: "Occurring"?

L287-290: The phrase "plays an important role" is used twice. please rephrase

L311: "carries"?

Response 2:

We agree This has been corrected.

Comments 3:

L285-297: Please add references

Response 3:

Supplemented with 4 items of references

Comments 4:

L285-297: Please add references

Response 4:

This passage has been rephrased

Reviewer 4 Report (New Reviewer)

Comments and Suggestions for Authors

The paper titled "Contents of macro- and microelements in blood serum and breast muscle of turkey subjected to pre-slaughter transport for various distances" addresses an important topic regarding the impact of pre-slaughter handling on the mineral content of turkey meat. The study found that pre-slaughter transport negatively affects the levels of macro- and microelements in turkey blood serum and breast muscle, with the most significant changes observed in the groups transported for longer distances (200 km and 300 km).

The key findings include:

Serum corticosterone levels were significantly higher in the 200 km and 300 km transported groups compared to the non-transported group, indicating increased stress response.

Blood serum of female turkeys transported 300 km had the lowest levels of most tested minerals, except for potassium which was highest in this group.

In male turkeys, pre-slaughter transport led to significant reductions in the levels of magnesium, phosphorus, calcium, iron, and sodium in the blood serum and breast muscle compared to the non-transported group.

The authors attribute the changes in mineral content to the physiological stress response and dehydration experienced by the birds during transport, which can impact nutrient metabolism and distribution.

Specific comments:

The title "Contents of macro- and microelements in blood serum and breast muscle of turkey subjected to pre-slaughter transport for various distances" is aligned with the manuscript content. It clearly conveys the main focus of the study, which is to assess the impact of pre-slaughter transport on the mineral composition of turkey blood serum and breast muscle. The title is complete and does not require any improvements.

The simple summary provides a concise overview of the study, highlighting the key points. It aligns well with the author's guidelines and does not require any suggestions for improvement.

The abstract accurately summarizes the study's objectives, methods, and main findings. It is correlated with the manuscript content and provides a clear and concise overview of the research. No suggestions for improvement are needed.

The keywords "turkey, stress, pre-slaughter handling, mineral components macro- and microelements" are relevant to the study and are present in the title. They adequately capture the main aspects of the research.

The Introduction provides sufficient background information to introduce the topic and justify the need for the study. It discusses the importance of pre-slaughter handling and its potential impact on poultry meat quality, particularly in terms of mineral content. However, I suggest including in the introduction a comparison of the published effects of the transport to the blood parameters and meat quality of other species, such as the pig, please include this information and see and cite: 10.3390/ani10122386 and 10.3390/ani10060945.

The Ethical statement is present and correct. The authors obtained permission from the Local Ethical Commission to conduct the study.

The Methods section provides a detailed description of the experimental design, including the pre-slaughter handling procedures, sample collection, and chemical analyses. The procedures are described with sufficient detail to allow for replicability of the study. No major suggestions for improvement are needed.

The sample size of 30 turkeys per group (male and female) seems adequate for the study. However, the authors do not report the sample size calculation or the statistical test used to determine the appropriate sample size.

The statistical analysis is correctly reported, with the use of means (xÌ„) and standard deviations (s) for each treatment group. The significance levels (P ≤ 0.01 and P ≤ 0.05) are also provided. No suggestions for improvement are needed.

The Discussion section starts by reiterating the aim of the study, which is to determine the content of macro- and microelements in the blood serum and breast muscle of turkeys subjected to different pre-slaughter transport distances.

The Discussion is comprehensive and provides a thorough interpretation of the results in the context of existing knowledge. Relevant references are provided to support the findings and discussions. No major suggestions for improvement are needed.

The authors discuss the practical implications of their findings, highlighting the negative impact of pre-slaughter transport on the mineral content of turkey meat and its potential consequences for meat quality.

The authors do not explicitly discuss the economic implications of their findings. However, they mention that changes in mineral content due to pre-slaughter stress can reduce meat quality, which may have economic consequences for poultry producers.

The authors do not specifically address the educational implications or perspectives of their findings in the manuscript. I recommend incorporating a discussion paragraph highlighting the significance of educating future veterinarians, technicians, and farmers about the issues addressed in the paper. Emphasizing the importance of effective teaching methods in shaping knowledgeable students and proficient veterinarians would add depth to the paper's implications. It is advisable to refer to recent publications on veterinary education to provide up-to-date insights into best practices in preparing future professionals to address the challenges discussed in the paper. Please see: 10.1016/j.jevs.2023.104537 and 10.3390/ani13223503.

The Conclusions are consistent with the evidence presented in the study and address the main objective of assessing the impact of pre-slaughter transport on the mineral content of turkey blood serum and breast muscle.

The references are properly cited throughout the manuscript and are relevant to the topic. The reference list includes appropriate and up-to-date sources.

Round 2

Reviewer 1 Report (Previous Reviewer 1)

Comments and Suggestions for Authors

Dear authors,

I'm sorry, but I have to state that the new version prepared by you is of worse quality than the previous one. Stylization and word order do not correspond to the higher demands placed on academic articles. That's why I recommend using Proofreadforme or an experienced long-time publishing professional for final proofreading. In this form, the article cannot be published and needs to be revised.

Comments on the Quality of English Language

Author Response

See attachment

Reviewer 3 Report (Previous Reviewer 3)

Comments and Suggestions for Authors

Authors made the majority of the corrections. However, in my opinion, the use of superscripts is still not appropriate in Tables 2, 3, 4 and 5 (superscripts are necessary for the whole row). Please ask advice by a statistician

Comments on the Quality of English Language

Minor editing of English language required

Author Response

See attachment

This manuscript is a resubmission of an earlier submission. The following is a list of the peer review reports and author responses from that submission.

Round 1

Reviewer 1 Report

Comments and Suggestions for Authors

The manuscript is written in good simple style, but I think that some parameters may increase its quality. I mention them in my review.

Reviewer 2 Report

Comments and Suggestions for Authors

Comments as attached

Reviewer 3 Report

Comments and Suggestions for Authors

Manuscript animals-2936906, entitled “Contents of macro- and microelements in blood serum and breast muscle of turkey subjected to pre-slaughter transport for various distances

Recommendation:       The above paper is not suitable for publication in its present form.

The article provides useful information about the effects of pre-slaughter transport distances on contents of macro- and micro- elements in blood serum and breast muscle in turkey. My main concern is that authors did not use superscripts appropriately in Tables, so it is difficult to understand which difference is significant. Please use superscripts in the entire row. At the same time, the part of discussion is poor.

L16: “feed” instead of “food”

L17: “…conditions can lead to unfavorable alterations in the…”

L30: Positively or negatively?

L31: “…microelements, both in the blood serum…”

L32: “due to the” instead of “in terms of”

L36: “…farmed poultry species. Due…”

L44-45: “may be lost”?

L49: Please delete “this”

L62-63: “…at 18-22 week. The female and male turkeys of the present study originated from one herd…”

L82: “withdrawn” instead of “taken”

L83: “…birds had only access…”

L84: “assigned” instead of “divided”

L84-85: Please delete “the birds 84 in the pre-slaughter procedure”

L90: “…around 1:00 p.m. (T-200 group).” What about T-300 group?

L94: Please delete “their”

L96: Laboratory or commercial conditions?

L98-99: “After chilling, samples (ca. 150 g) from the right breast muscle were collected for laboratory analyses. Laboratory analyses of blood…”

L117: “of broiler chickens”?

L124: “variation were” instead of “differentiation was”

L126: “the examined minerals” instead of “tested components” Please refer only to significant differences. What about calcium?

L127: Is this difference significant?

L132: “In his case”?

L135: Please rephrase

L142: “to 158.53”? mmol or μmol?

L176: “The blood of female turkeys…”

L232-234: Too general. Not in all cases (not in females)

L246: What about the 300km?

L258: “a tan elevated temperature”?

L262: Positively or negatively?

Comments on the Quality of English Language

Moderate editing of English language required